# Hopfions emerge in ferroelectrics

I. Luk'yanchuk[1], Y. Tikhonov[1,2], A. Razumnaya [2] & V. M. Vinokur [3✉]

Paradigmatic knotted solitons, Hopfions, that are characterized by topological Hopf invariant, attract an intense attention in the diverse areas of physics ranging from high-energy physics, cosmology and astrophysics to biology, magneto- and hydrodynamics and condensed matter physics. Yet, while being of broad interest, they remain elusive and under-explored. Here we demonstrate that Hopfions emerge as a basic configuration of polarization field in confined ferroelectric nanoparticles. Our findings establish that Hopfions are of fundamental importance for the electromagnetic behavior of the nanocomposits and can result in advanced functionalities of these materials.

[1] Laboratory of Condensed Matter Physics, University of Picardie, 80039 Amiens, France. [2] Faculty of Physics, Southern Federal University, 5 Zorge str., Rostov-on-Don, Russia 344090. [3] Materials Science Division, Argonne National Laboratory, 9700 S. Cass Avenue, Argonne, IL 60637, USA. ✉email: vinokour@anl.gov

Confinement of a ferroelectric material changes its electric properties radically. Termination of the of polarization at the surface leads to the depolarization charges $\rho = -\text{div } \mathbf{P}$ that produce the depolarization field $\mathbf{E}_d$. In turn, the self-consistent interactions result in a nonuniform texture that minimizes electrostatic energy costs associated with these depolarization effects. The corresponding topologically nontrivial textures include regular patterns of Kittel domains[1–5], which can be also viewed as the periodic array of vortex–antivortex pairs[6] and lattice of skyrmions[7] in the films and superlattices, vortices and skyrmions in the nanords, nanodots[8–15], and nanoparticles[16,17]. As we show below, a geometrical restriction of a ferroelectric brings about a fundamental class of the topological formations, Hopfions, that appear inherent to a broad variety of nature phenomena[18–29].

## Results

**Hopfion topology and Arnold theorem.** We consider a spherical nanoparticle described by the isotropic free energy functional. It is an exemplary system capturing the global topological properties of a confined ferroelectric nanodot. After that we introduce the perturbations, including anisotropy and electrostrictive coupling, which preserve the topological stability of the solution. A uniform mono-domain state of the nanoparticle is not energetically stable because of formation of the surface depolarization charges located at the termination points of polarization lines $\mathbf{P}$, see Fig. 1a. To minimize the energy associated with the depolarization field, $\mathbf{E}_d = -\mathbf{P}/\varepsilon_0$ (where $\varepsilon_0 = 8.85 \times 10^{-12}\,\mathrm{C\,V^{-1}\,m^{-1}}$), the system transforms itself into a structure with the vanishing depolarization charges so that $\text{div } \mathbf{P} = 0$. Therefore, the divergenceless of the polarization field is the fundamental condition defining the physics of the spatially nonuniform ferroelectricity. The absence of the depolarization charges at the surface, implies that the polarization vector, $\mathbf{P}$ is tangent to the surface of the particle.

An instant configuration stemming from the above conditions is the vortex[11], see Fig. 1b. For the case of the isotropic spherical nanoparticle, such a solution[30] is stable just below the transition from the high-temperature paraelectric phase into the ferroelectric phase. However, in general, far from the transition, the system seeks for the configuration in which the amplitude of the polarization remains close to its equilibrium value everywhere, hence strives to eliminate singularities.

A singularity at the vortex core can be removed by the continuous deformation of the vector field $\mathbf{P}$ promoting its escape into the third dimension along the vortex axis[31], see Fig. 1c. Had this process been occurring in an unrestricted 3D space, it would have resulted in a uniform polarization. However, in the confined spherical geometry, this would have recovered the unfavorable mono-domain configuration shown in Fig. 1a. To avoid that, the polarization flow along the vortex axis spreads into a back-flow over the sphere's surface, maintaining polarization tangent to the surface hence avoiding the onset of depolarization charges. The resulting $\mathbf{P}$-field configuration is a 3D knotted soliton, called "Hopfion," which is a set of interlinked circles or torus knots, see ref. [24] and references therein. A simplest single polarization Hopfion is shown in Fig. 1d.

To unravel the nature of an emergent polarization structure in a nanoparticle, we observe that the lines of the divergenceless polarization field, $\mathbf{P(r)}$, have no intersections, are looped, tangent to the surface of the nanoparticle, and constitute a dense set in the sphere. Polarization lines are identical by their topological characteristics to the streamlines of an ideal incompressible liquid inside the restricted volume, which enables us to employ topological methods of hydrodynamics developed by Arnold[32]. According to Arnold's theorem, the analytic stationary flows of the divergenceless vector field can belong in either of two classes: the field flows fibered into the nested cylindrical surfaces and those fibered into the set of the nested tori. The former configurations correspond to vortices and the latter configurations constitute Hopfion. We describe a layout of a Hopfion starting with filling the interior of the nanoparticle by the set of sequentially nested concentric tori, as shown in Fig. 2a. Then polarization field lines form the dense set of trajectories twining these tori, as shown by thin solid lines in Fig. 2b. A topological structure that implements the streamlines of Hopfion is characterized by its topological charge called Hopf invariant[32]

$$\mathcal{H} = \int_{\mathbb{M}^3} (\mathbf{P} \cdot \mathbf{A}) d\mathbf{r}, \tag{1}$$

where the gauge field $\mathbf{A}$ is defined by $\mathbf{P} = \text{rot} \mathbf{A}$. This extends the concept of an integer Hopf charge of the fields representing the trace of the unit vector like spin in magnetic systems[24,27,28] or director in liquid crystals[29] maintaining its direction, onto the wider class of the confined divergenceless fields. Importantly, the Hopf invariant generalized by Arnold, is not necessary integer but still conserves under the action of an arbitrary volume-preserving diffeomorphism so that the Hopfion is stable and does not relax out. The definitive property of the polarization lines in a Hopfion is that each field line links through others. This property illustrates an equivalent topological definition of the Hopf invariant. According to Arnold[32], the average self-linking (not necessarily integer) of a confined divergenceless vector field coincides with $\mathcal{H}$. Therefore, the observed non-zero linking is a fingerprint of the Hopf fibration. However, the link indices of different polarization loops can vary, and the exact value of $\mathcal{H}$ is obtained only upon averaging.

**Chirality of the Hopfion state.** An associated feature that arises in the Hopfion state is the chirality, which is the asymmetry with respect to mirror-reflection. The corresponding symmetry group is $C_\infty$, and is eventually reduced by the crystal anisotropy. Hopfions can be the "left" and "right" ones, hence spontaneous chiral symmetry breaking upon the formation of the Hopfion state. Chirality marks the Hopfion off from the vortex, endowed with the group $C_{\infty h}$ that includes the reflection in the plane, $\sigma_h$,

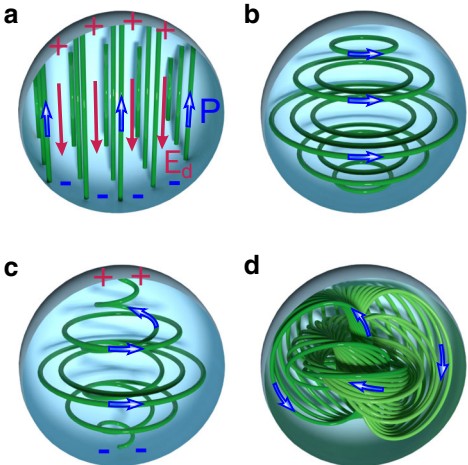

**Fig. 1 Formation of the Hopfion. a** A uniform distribution of the polarization, **P**, (green lines) in a spherical nanoparticle, blue arrows showing polarization direction. Positive and negative depolarization charges on the surface induce depolarization electric field, **E_d** (red lines). **b** Polarization vortex. **c** Escape of the polarization vortex into the third dimension. **d** Polarization Hopfion.

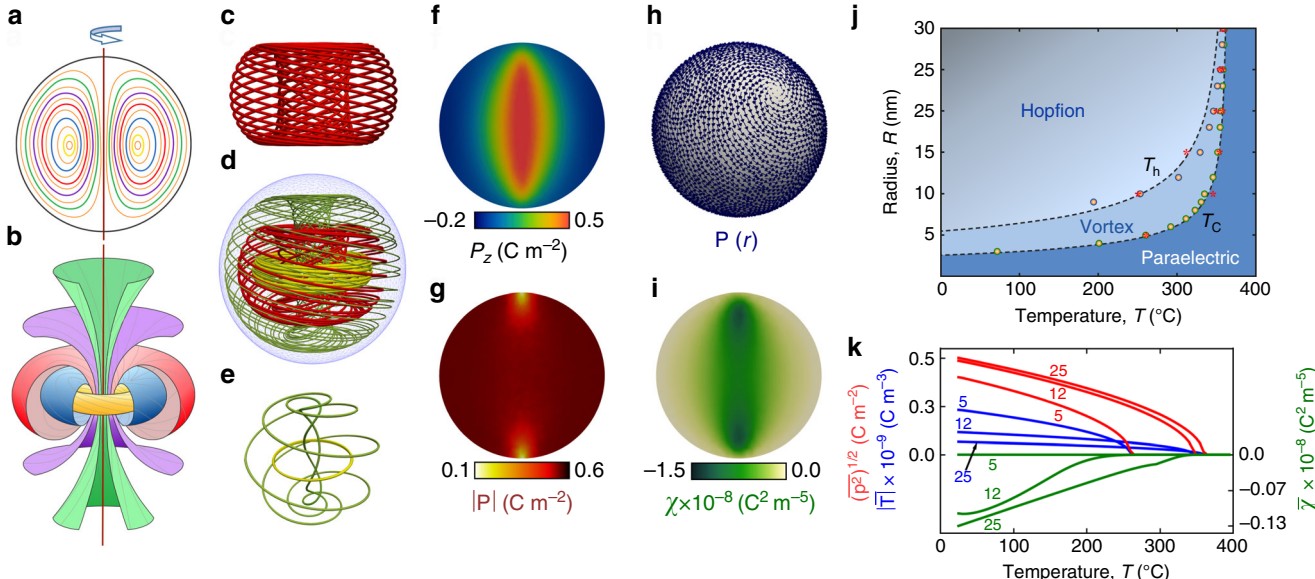

**Fig. 2 The Hopfion structure. a** A cross section of the spherical nanoparticle exposing a cut across the set of nested concentric tori. **b** The set of tori obtained by rotation of the loops shown in **a** around the vertical axis. Each torus is entwined by a dense set of the polarization lines[49]. **c** A single Hopfion torus wrapped up by polarization lines calculated in the framework of the isotropic model. **d** Three nested tori entwined by the polarization lines illustrating their tendency to densely fill up the nanoparticle. **e** Exemplary polarization lines belonging in the tori shown in **d** that illustrate linking of the loops in the Hopfion. **f** The color map of the $P_z$−component of the polarization over the meridian cross section of the nanoparticle. **g**, The color map of the distribution of the amplitude of the polarization, |**P**|. **h** Whirlpool distribution of the polarization over the surface of the nanoparticle near the Hopfion axis. **i**, The color map of the chirality, $\chi$, distribution inside the nanoparticle. **j** The temperature-radius, $T$–$R$, phase diagram of the Hopfion and vortex topological states in the isotropic spherical nanoparticle. The circles display results of numerical computations of the paraelectric-vortex, $T_c$, and vortex-Hopfion, $T_h$, transition temperatures. The dashed lines correspond to the geometrical scaling fit, see text. Importantly, the vortex phase occupies a very narrow region in the vicinity of the paraelectric-ferroelectric transition and the dominant part of the phase diagram corresponds to the Hopfion state. The color gradient reflects emergence of the chirality. The red asterisk symbols stand for the transitions in the anisotropic ferroelectric PZT nanoparticle. **k** The temperature dependencies of the ferroelectric characteristics, mean-squared polarization, $(\overline{\mathbf{P}^2})^{1/2}$, (blue lines), the absolute value of the mean toroidal moment, $|\overline{\mathbf{T}}|$, (red lines), and the mean chirality $\overline{\chi}$ (green lines), taken for three representative nanoparticle radii, $R$ = 5, 12, and 25 nm.

perpendicular to the vortex axis. We thus use the chirality parameter, $\chi = \mathbf{P} \cdot \mathrm{rot}\mathbf{P}$[16], to characterize the Hopfion state. This parameter compliments the toroidal moment $\mathbf{T} = \mathrm{rot}\,\mathbf{P}$ that is ordinarily used for the description of the state containing topological excitations in ferroelectric nanoparticles[8], since $\mathbf{T}$ cannot expose the difference between the vortex state and the chiral Hopfion state. Note, that the spontaneously arising chirality opens an opportunity of manipulating the ferroelectric nanoparticles by circular-polarized laser tweezers, inducing and tuning the optical activity of the media.

**Hopfion structure in a ferroelectric**. To investigate the Hopfions arising in ferroelectric nanoparticle, we perform the relaxation minimization of the Ginzburg–Landau (GL) functional coupled with the electrostatic and elastic equations, see "Methods" and Supplementary Methods section in Supplementary Information (SI). An insight into the Hopfion emergence is gained by the purposeful initial using the isotropic model functional capturing, as mentioned above, the global topological properties of a nanodot. We select, however, parameters that are close to those in the realistic oxide materials and that partially account for the elastic interaction. Shown in Fig. 2c–k are the results of computations. Figure 2c, d display the Hopf fibration in the isotropic nanoparticle with radius $R = 25$ nm at room temperature realizing the self-linking spiral-like structure of polarization lines. The dense set of lines forming the knots at a single torus is shown in Fig. 2c, whereas the panel Fig. 2d exhibits the compactification of entwined tori in the bulk of the nanoparticle. Figure 2e demonstrates the pairwise linking of the polarization lines belonging to the tori shown in Fig. 2d. The nontrivial knotting of the field lines

leads to the peculiar spatial distributions of the polarization characteristics of the system. The tendency for the polarization vectors to escape in the third dimension results in the up-stream of the polarization lines near the Hopfion core and to their downstream at the periphery, as reflected in Fig. 2f showing the $P_z$ component. At the same time, the distribution of amplitudes of polarization vectors, becomes nearly homogeneous, (Fig. 2g), and the residual singularities settle as whorles of the polarization at the points of the termination of the Hopfion axis at the poles of the sphere as shown in Fig. 2h. These residual singularities are essentially non-removable and manifest the Poincaré hairy ball theorem stating that there is no non-vanishing continuous tangent vector field on two-dimensional sphere[33]. Fig. 2i demonstrates the distribution of the chirality, $\chi$, inside the particle that concentrates mostly along the Hopfion core.

Panel (j) presents the $T$–$R$ phase diagram for the spherical particles with the radius $R < 30$ nm. Notably, the Hopfion state occupies its major part. The transition temperature $T_c$ from the high-temperature paraelectric state to the low-temperature ferroelectric one, lies only slightly below the bulk temperature $T_0$ in large particles with $R > 20$ nm. In small particles with $R < 20$ nm, $T_c$ is noticeably suppressed by the size-driven confinement. The polarization texture of the ferroelectric state, which forms just below the transition, has the vortex-like structure. In general, the dependence $T_c(R)$ is well fitted by the formula, following from the dimensional analysis of GL equations, $(T_0 - T_c)/T_0 \simeq (\mu \xi_0/R)^2$, where $\xi_0 \simeq 0.7$ nm is the coherence lengths and $\mu_c \simeq 2.0$. Vortices start expelling their core singularities into the third dimension below the critical temperature $T_h$, which also scales as $R^{-2}$, with the coefficient $\mu_h \simeq 2.8$. The temperature interval of the vortex

phase existence is negligibly small for $R > 10$–$15$ nm and further cooling drives the system into a Hopfion state. The vortex state becomes noticeable only in small enough nanoparticles, where the geometry restriction stabilizes vortices.

The temperature dependence of the principal ferroelectric characteristics, the mean-squared polarization, $(\overline{\mathbf{P}^2})^{1/2}$, the absolute value of the mean toroidal moment, $|\overline{\mathbf{T}}|$, directed along the Hopfion/vortex axis, and the mean chirality, $\overline{\chi}$, chosen negative for concreteness, are shown in the Fig. 2j for three characteristic sizes of the nanoparticles: $R = 5$, 10, and 25 nm. The mean-squared polarization vanishes as a square root on approach to the ferroelectric transition temperature, $T_c$, similar to the uniform bulk case. The toroidal moment also vanishes at $T_c$, whereas the chirality disappears below $T_c$ at the vortex-Hopfion transition and is used to determine $T_h$. Note that for the 5 nm nanoparticles $\overline{\chi} = 0$ since the system remains in the vortex state.

**Hopfion state in a PbZr$_{0.6}$Ti$_{0.4}$O$_3$ nanoparticle and its field dependence.** Now we are equipped to turn to concrete embodiments of Hopfions in specific ferroelectric materials. The topological stability of Hopfions is secured by the Kolmogorov–Arnold–Moser (KAM) theorem[34], generalized for the three-dimensional systems with static divergenceless (volume preserving) vector fields[35,36], stating that the tori winding of the vector field remains intact under the adiabatic nonlinear perturbations of the free energy. In our case, the perturbations are the anisotropy and the electrostrictive coupling that deform Hopfions but do not destroy them. We focus on the quasi-isotropic PbZr$_{0.6}$Ti$_{0.4}$O$_3$ (PZT) compound, which is close to the so-called morphotropic phase boundary. Numerical simulations of the PZT nanoparticle of $R = 25$ nm are based on the relaxation of the full GL functional with the elastic terms, see "Methods" and SI. We find that the Hopfion state spans almost the entire ferroelectric phase, so that the resulting phase diagram is close to the phase diagram for the isotropic system with the similar parameters, as evidenced by the calculated points of transitions in PZT, shown by asterisks in Fig. 2j. The field-polarization characteristics of the nanoparticle of $R = 25$ nm at room temperature are shown in Fig. 3. The distinct polarization topological configurations are marked as states (i)–(vii). The set-up used in the simulations of the nanoparticle under the applied field is sketched in the insert in the plot. The voltage $U$ is applied to the plates of the capacitor embracing the nanoparticle, which is oriented to have its [111] crystallographic axis perpendicular to the capacitor's plates. Accordingly, $U$, induces average polarization, $\overline{P}$, and average internal field, $\overline{E}$, along the [111] direction. The relation between these quantities completely describes the dielectric properties of the system, and is the constitutive relation $\overline{P}(\overline{E})$. We show that $\overline{P}(\overline{E})$ dependence is the S-shape curve with slight hysteresis that we describe below.

The state (i) in Fig. 3 corresponds to the Hopf fibration arising after the zero-field quench of the nanoparticle from the paraelectric state. As shown in Supplementary Figs. 1 and 2 in SI, this configuration maintains the torus-winding character but the torus now is deformed by the crystal anisotropy which bends the polarization lines toward the equivalent to [111] crystallographic directions, which in PbZr$_{0.6}$Ti$_{0.4}$O$_3$ corresponds to the minimal anisotropy energy, and fixes the Hopfion axis along [111]. The resulting polarization texture is now alike the 6-fold flux-closure structure of the adjoining domains. The anisotropy-induced depolarization charges and fields remain, however, vanishing small. In addition, the pole singularities extend into the bulk, to form two respective vortices coexisting with the bulk Hopfion.

We next investigate the effect of the possible presence of the semiconducting or impurity-induced free charges that can cause

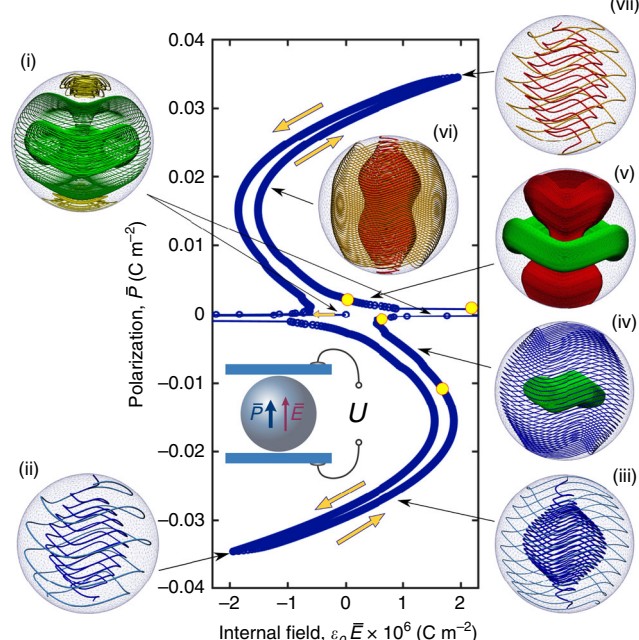

**Fig. 3 Field-induced topological states in PbZr$_{0.6}$Ti$_{0.4}$O$_3$ (PZT).** The $\overline{P} - \overline{E}$ polarization curve (blue line) demonstrates hysteretic behavior due to the series of transitions (denoted by yellow circles) between different polarization topological states (i-vii) in the 25 nm nanoparticles of the PZT. The helices with the "upward" and "downward" polarization flux are shown in brick-red and blue, respectively, the green colour depicts the localized Hopfion polarization lines, and yellow denotes the closed vortices. The block yellow arrows show the direction of the evolution along the hysteresis loop. The inset shows a conceptual setup. The applied voltage, $U$, induces average polarization, $\overline{P}$, and average internal field, $\overline{E}$. The nanoparticle is oriented to have its [111] crystallographic axis perpendicular to the capacitor's plates.

electric screening. To that end we add the terms accounting for the screening contribution in the GL functional (see "Methods"). At typical for the PZT values of the screening length about $\lambda = 80$–$100$ nm[37,38], screening does not influence the Hopfion texture. At elevated density of the free charges where $\lambda \simeq 20$ nm the polarization lines maintain the winding texture. However, the arising volume and surface charges break down the condition for the field be divergenceless hence the polarization lines can now thrust the surface of the nanoparticle. Accordingly, this violates the conditions for the Arnold's theorem, the fibered nested tori structure slightly distorts, and the tori-twinning polarization lines convolve or untwine along the spiral-like paths and creep from torus to torus, see Supplementary Fig. 3 in SI. Further increasing free charge density so that $\lambda \lesssim 5$ nm cuts Hopfions and the cells harboring vortex-like textures emerge. The complete unwinding of the polarization lines into the monodomain structure requires the very high metal-like concentration of the free carriers with $\lambda < 0.05$ nm (see SI). Note, however, that the strong anisotropy of the sample (natural or strain-induced) can restitute the bound charges in a form of head-to-head or tail-to-tail charged domain walls[39], distributed depolarization charges in the soft Kittel domains[4], or in polar topological defects[40] when the anisotropic energy becomes comparable with the screened electrostatic one.

Upon applying the external field (in the negative direction), the virgin curve $\overline{P}(\overline{E})$ jumps first to the left, across the singular internal field, $\varepsilon_0\overline{E} \simeq 1.4 \times 10^{-5}$ C m$^{-2}$, related to the topological piercing of the Hopfion by helical polarization lines that will be

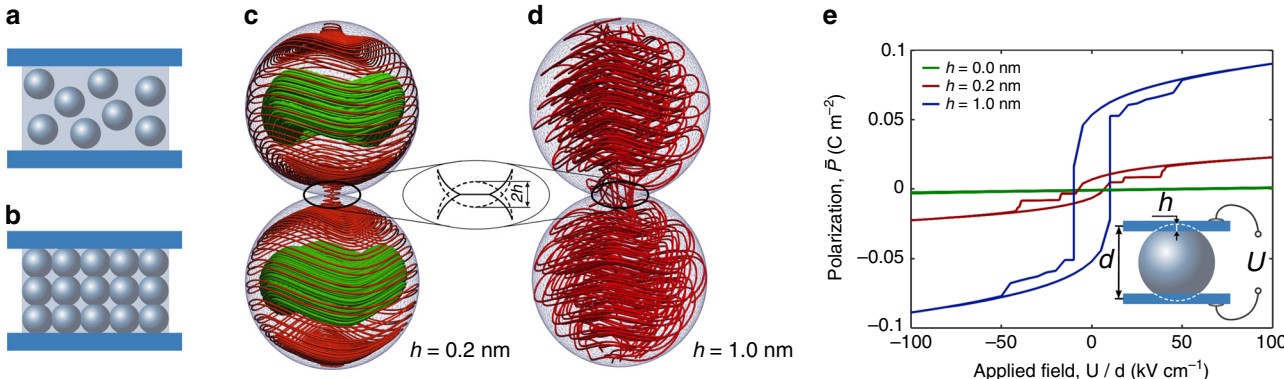

**Fig. 4 Polarization in a nanoparticle composite. a** Dilute composite of isolated nanoparticles in a capacitor. **b** Sintered composite of contacting nanoparticles. **c** Field lines flow between moderately densified nanoparticles of $R = 25$ nm, contacted along the [111] direction. The densification degree is characterized by the thickness of the connecting neck $2h = 0.4$ nm, see inset. The delocalized helical lines flowing between the particles are embraced by the localized lines forming the toroidal Hopfion states. **d** In a composite of strongly densified nanoparticles with $2h = 2$ nm, all the field lines form delocalized helices flowing across the entire composite. **e** The polarization hysteresis loops for a single nanoparticle in the sintered composite as a function of the applied field, $U/d$ for different densification degrees $h$. The inset sketches a model particle with cut skullcaps of the height $h$ confined between the electrodes which mimics the behavior of the composite.

described further. Then it descends along the left branch of the hysteresis loop forming finally the down-oriented directional helical structure (state (ii) in Fig. 3). In the emerging polarization structure, open polarization lines thrust the nanoparticle, so that the mean polarization flux gets aligned with the applied field.

Reversing the change in the field, we follow now the right hand side branch of the hysteresis loop from the bottom to the top. The evolution of the system upon the monotonic variation of the field from the negative to the positive value, occurs through first, compression, and then, stretching the helical polarization lines with the change of the mean polarization flux direction from the negative to the positive one. We observe that the system passes through the sequence of topological phases that follow the Arnold's partitioning of the nanoparticle space into cells where the polarization lines are entwined around the nested sets of either cylindrical or toroidal surfaces, see in Supplementary Fig. 2 in SI for details. In the initial helical state (ii) the polarization lines are entwined around the cylindrical surfaces. Upon the decreasing field, the helical structure compresses while broadening its central part (state (iii)) and, finally, a spherical cell containing the toroidal Hopf fibration nucleates at the center of the nanoparticle (state (iv)). The emerging Hopfion grows further ousting the helical states toward the nanoparticle periphery, which bypass Hopfion outside and carry the mean polarization of the nanoparticle along the nanoparticle surface. When the applied field vanishes, the Hopfion fills up the entire nanoparticle asymptotically approaching to the state (i) with $\bar{P} = 0$.

The change of the sign of the external field from negative to positive leads to the topological phase transition in the course of which the system "turns inside out" and a hyperbolic cell filled up with the nested cylinders sets in along the axis of the nanoparticle. This cell ruptures the Hopfion sphere into a torus (state (v)) and hosts the open polarization helical lines carrying now the mean polarization inside the nanoparticle along its axis. This transition occurs abruptly at some threshold applied field and manifests as a disruption of the smooth behaviour of the internal field $\bar{E}$ which makes a singular turn in polarization curve. Upon the further field increase, the Hopfion torus retires toward the equator and eventually disappears there draining out of the nanoparticle, which thus falls into the helical state. Just after the transition the polarization helices maintains the fitted structure (state (vi)), as a legacy of the vanished Hopfion tori, and only at higher applied fields it crosses over to the helical state (vii) equivalent, up to sign

$P(r)$ reversal, to the state (ii). Upon the sequential field reversal and decreasing the field back, the system does not pass through the reverse sequence of the states, but repeats the (ii) → (vii) scenario with the replacement $P(r) \to -P(r)$, demonstrating thus the weakly hysteretic behaviour (left descending branch in Fig. 3). Again, the mean polarization flows, first, along the surface of the nanoparticle and then along the nanoparticle axis.

**Discussion**

Complexity of the intertwined topological states encoded in the $\bar{P} - \bar{E}$ characteristic of Fig. 3 stems from the interplay of confinement and depolarizing effects. Most importantly, the system is highly responsive to even weak internal fields, owing to the utmost softness of the helical springs of the polarization lines. In other words, the ease at which the open lines reconnect, ensures an unobstructed redistribution of the field-induced depolarization charge at the points of their termination, guaranteeing the almost perfect screening of the applied field. This behaviour is similar to that of the ferroelectric with domains, where the easy domain wall motion results in the similar softness[41]. Because of that, the absolute value of the effective dielectric permittivity of the nanoparticle, $\bar{\varepsilon} = \bar{P}/\varepsilon_0 \bar{E}$, can reach giant values of order $10^4$ and even more. Moreover, in the close resemblance of the nanoparticles with domains, the S-shape $\bar{P}-\bar{E}$ characteristic demonstrate segments having the negative slops, hence negative capacitance effect, which is explained by the advancing reaction of the polarization texture to the applied field[42]. This $\bar{P}-\bar{E}$ characteristic and, in particular, the negative value of $\bar{\varepsilon}$ can be extracted from the total capacitance of the measuring capacitor shown in the inset of Fig. 3 using the Maxwell Garnett mixing rule[43] for the nanoparticle in dielectric matrix.

Note further, that in larger particles, polarization lines maintain their anisotropic toroidal winding, but new small toroidal formations, "Hopfioninos," which can be perceived as newly nucleating domains, bud off from the original Hopfion body, in Supplementary Figure 4 in SI. Eventually, upon further increasing of the particle size, a Hopfionino array can develop into a chaotic texture.

We investigate now the Hopfion-governed physics in a composite material comprising the high-$\varepsilon$ ferroelectric nanoparticles embedded in the low-$\varepsilon$ dielectric matrix. Note, that according to the Maxwell Garnett mixing rule the dilute high-$\varepsilon$ nanoparticles, see Fig. 4a, only weakly renormalize the properties of the low-$\varepsilon$

hosting matrix. This was confirmed by numerical simulation in[44]. We thus focus on the topological excitations in composite sintered nanoparticles, where polarization lines can pierce the entire system passing from one grain to another. The emergent interconnected Hopfion-like excitations in disordered ensemble of contacting ferroelectric nanoparticles can also serve as a good model of the polar nanoclusters in relaxors. For now, however, in order to gain an insight into the behavior of sintered materials, we restrict ourselves to a a rectangular array of contacting nanoparticles as shown in Fig. 4b.

Figure 4c displays the configuration that forms under the conditions of the moderate densification of PZT nanoparticles with $R = 25$ nm, contacted along the [111] direction. The degree of densification is quantified by the thickness of the contacting neck, $2h = 0.4$ nm (see inset), so that the area of the mutual contact of the adjacent particles serves as an aperture for polarization lines. Only a fraction of polarization helical lines passes through the interfacial aperture. Another part of the lines is confined into the Hopfions and does not interact with the applied electric field. They are invisible to the entire dielectric response of the system. In the case of strongly densified nanoparticles with $2h = 2$ nm all the field lines form helices flowing through the area of the contact (Fig. 4d).

Figure 4e displays the polarization characteristics of the nanocomposite, comprising the sintered 25 nm nanoparticles, as functions of the applied voltage. The chain of the connected particles with the repeating polarization pattern is modeled by a single particle with cut off the skullcaps of the height $h$, the electrodes covering its top and bottom cuts respectively. The model setup is shown in the inset. As we already mentioned, the particles with the vanishing contact area ($h \simeq 0$ nm) give almost no contribution to the dielectric properties of the system. The dielectric response grows with the degree of the densification. For the moderate contact $h = 0.2$ nm the switching between the up- to down-polarized helices, occurs along the gently sloping hysteresis loop. At each branch, the system quasistatically passes through the sequence of the topological states, similar to those, shown in Fig. 3. For the high degree of the densification, $h = 1$ nm, the sharp hysteresis loop with the polarization jumps is observed. The switching follows through the same sequence of the states, but the system passes through the Hopfion states via dynamic instability, as illustrated in Supplementary Movie 1.

## Methods

**Functional**. The free functional, $F$, is written as follows

$$F = \int \left( \left[ a_i(T)P_i^2 + a_{ij}^u P_i^2 P_j^2 + a_{ijk}P_i^2 P_j^2 P_k^2 \right]_{i \le j \le k} \right.$$
$$+ \frac{1}{2} G_{ijkl}(\partial_i P_j)(\partial_k P_l) + (\partial_i \varphi)P_i - \frac{1}{2}\varepsilon_0 \varepsilon_b \left[ (\nabla \varphi)^2 + \lambda^{-2}\varphi^2 \right]$$
$$\left. - C_{ijkl}Q_{klmn}u_{ij}P_m P_n + \frac{1}{2}C_{ijkl}u_{ij}u_{kl} \right) d^3 r , \qquad (2)$$

were the summation over the repeated indices $i, j, \ldots = 1, 2, 3$ (or $x, y, z$) is performed. The first square brackets term of Eq. (2) stands for the Ginzburg–Landau energy written in the form given in ref. [45]. Importantly, the 4th-order coefficients $a_{11}^u$ and $a_{12}^u$ (and their cubic-symmetry homologs) are taken at zero strain and are calculated by the Legendre transformation from the stress-free coefficients $a_{11}^\sigma, a_{12}^\sigma$ [46], see also ref. [47]. The second term of Eq. (2) with coefficients $G_{ijkl}$ corresponds to the gradient energy. The last terms are the electrostatic and elastic energies, written in terms of the electrostatic potential $\varphi$ and strain tensor $u_{ij}$, respectively. Here $\varepsilon_0 = 8.85 \times 10^{-12}$ CV$^{-1}$ m$^{-1}$ is the vacuum permittivity, $\varepsilon_b \simeq 10$ is the background dielectric constant of the non-polar ions, typical for PbTiO$_3$[16], $\lambda$ is the screening length due to the free carriers when present, $C_{ijkl}$ is the elastic stiffness tensor and $Q_{ijkl}$ is the tensor of electrostrictive coefficients.

The polarization-induced distribution of the electrostatic potential $\varphi$ and elastic strains $u_{ij}$ in functional Eq. (2) are found as solutions of equations

$$\varepsilon_0 \varepsilon_b [\nabla^2 - \lambda^{-2}]\varphi = \partial_i P_i , \qquad (3)$$

$$C_{ijkl}\partial_j (u_{kl} - Q_{klmn}P_m P_n) = 0 . \qquad (4)$$

**Material parameters**. For the PbZr$_{0.6}$Ti$_{0.4}$O$_3$ the coefficients for the uniform part of the functional Eq. (2) are as follows, $a_1 = 2.3$ $(T - 364 \text{ °C}) \times 10^5$ C$^{-2}$ m$^2$ N, $a_{11}^u = 0.44 \times 10^9$ C$^{-4}$ m$^6$ N, $a_{12}^u = 0.074 \times 10^9$ C$^{-4}$ m$^6$ N, $a_{111} = 0.27 \times 10^9$ C$^{-6}$ m$^{10}$ N, $a_{112} = 1.21 \times 10^9$ C$^{-6}$ m$^{10}$ N, and $a_{123} = -5.69 \times 10^9$ C$^{-6}$ m$^{10}$ N. The electrostriction coefficients are $Q_{1111} = 0.073$ C$^{-2}$ m$^4$, $Q_{1122} = -0.027$ C$^{-2}$ m$^4$, and $Q_{1212} = 0.016$ C$^{-2}$ m$^4$ (with cubic symmetry permutations) were calculated on the base of the expression, given in ref. [46] after transformation from Voigt to tensor notations[47]. The gradient energy coefficients $G_{1111} = 2.77 \times 10^{-10}$ C$^{-2}$ m4 N, $G_{1122} = 0$, and $G_{1212} = 1.38 \times 10^{-10}$ C$^{-2}$ m$^4$ N selected to be the same as for PbTiO$_3$[48]. The elements $C_{1111} = 1.68 \times 10^{11}$ m$^{-2}$N, $C_{1122} = 0.82 \times 10^{11}$ m$^{-2}$ N, and $C_{1212} = 0.41 \times 10^{11}$ m$^{-2}$ N of the stiffness tensor $C_{ijkl}$ were obtained by inversion of the compliance tensor $s_{ijkl}$, which elements are given in ref. [46]. To evaluate the effect of screening we varied the screening length from $\lambda = 80$–100 nm which is typical for the PZT[37,38] down to nanometers.

To explore the isotropic model we dropped out the elastic part of the functional Eq. (2), neglected the 6th-order polarization terms, and imposed $a_{12}^u = 2a_{11}^u = 0.27 \times 10^{-9}$ C$^{-4}$ m $^6$N, so that the uniform part of the functional acquired the isotropic form $a_1(T)P^2 + a_{11}^u P^4$. Note that the gradient energy with selected coefficients $G_{ijkl}$ is already invariant with respect to rotation.

## Data availability
Computational scripts are available online at https://github.com/ferroelectrics/hopfion.

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

## Acknowledgements

We are pleased to thank J. Mangeri, S. P., Nakhmanson, and O. Heinonen for illuminating discussions. This work was supported by the U.S. Department of Energy, Office of Science, Basic Energy Sciences, Materials Sciences and Engineering Division (V.M.V and partially I.L.) and by H2020 ITN-MANIC action (I.L. and Y.T.), by the Southern Federal University, Russia (A.R. and Y.T.).

## Author contributions

I.L., Y.T., A.R., and V.M.V. conceived the work and performed calculations. I.L. and V.M.V. wrote the paper.

## Competing interests

The authors declare no competing interests.
