## [Peer Review File · Nature Communications]

Editorial Note: This manuscript has been previously reviewed at another journal that is not operating a transparent peer review scheme. This document only contains reviewer comments and rebuttal letters for versions considered at Nature Communications .

Reviewers' comments:

Reviewer #1 (Remarks to the Author):

In the revised versions of the manuscript and supplemental information, the authors have presented additional simulation data as well as a sufficiently detailed description of the employed methods. Overall, I find that the quality of the presentation has significantly improved and that the authors have addressed almost all of my previous comments. In my opinion, the revised version is technically sound and does not raise any major questions.

At the same time, some minor points remain to be clarified. Specifically, I was not convinced by the response of the authors about the topological characterization of the Hopfion state. Specifically, I suggested that the authors should use Eq. (1) of the manuscript to confirm the Hopfion topology. In response, the authors have indicated that the gauge field A appearing in this formula does not have a physical meaning and that there is no common approach to computing the Hopf index. Why then did the authors provide this formula in the first place if they do not find it useful? At the same time, the Hopf index was already computed by several authors for spin patterns (see e.g. Ref. [28] provided in the manuscript). I do not see any reason why it could not be done for the case of polarization fields discussed in this manuscript. In case the authors do not wish to perform such calculations, I suggest that it is explicitly stated in the main text of the manuscript that the topological characterization is performed through visual inspection of the field patterns and the integer value of the visually assessed topological charge (or linking index) should be provided. The fact that authors did not compute the Hopf index is not a conceptual problem. However, visual inspection can leave certain facts unnoticed and I think it would be better if the method employed by authors to identify the topology of field patterns is stated in an unambiguous manner.

Moreover, I disagree with the statement "To conclude, changing the Hopfion by screening would require an unattainable high free charges density and, therefore, we can neglect screening in our calculations" that the authors have added at the end of the methods section. Specifically, I disagree with the use of the word "unattainable". Based on the arguments I have provided in the comment about the screening effects I do not think that such situation is unattainable. Existence of the head-to-head or tail-to-tail charged domain walls is one of the counterexamples. Another specific example (among many) is the distribution of bound charges in polar topological defects found in FE thin films (see Fig. S5(a) in Phys. Rev. Lett. 120,177601 (2018)). In my opinion, this formulation should be changed.

Finally, I would like to point out that in the revised version the authors make a reference to Kolmogorov-Arnold-Moser (KAM) theorem to explain why including elastic and electrostrictive free energy contributions does not alter the Hopfion topology. For me it is not obvious to see why this theorem should apply to the case of static polarization fields considered by the authors. The reference provided by the authors (Arnold's classical mechanics book) does not help in clarifying the situation since it is very general and provides many different formulations of the theorem. In my opinion the authors need to explicitly demonstrate how exactly the studied case fit either of these formulations.

In conclusion, I would like to mention once again that, despite some issues with specific formulations used by the authors, this study contains rather elegant ideas that would make it interesting not only for researchers working in the fields of ferroelectrics and related materials.

Dr. S. Prokhorenko

Reviewer #2 (Remarks to the Author):

This review is for a revised version of the authors original manuscript previously submitted to Nature Physics. I am pleased to see that the author's revised manuscript contains additional clarifying information in the main text and supplementary information, which helps to give a more complete picture for these new observations, and I think that the work is in a publishable state. In their rebuttal, the author's also make a good case that their first-time prediction of the ferroelectric hopfion will likely draw attention from the theoretical community within ferroelectrics and the wider community working on hopfion theory. Although the study is an interesting and original work, it is my opinion that this study is unlikely to trigger significant additional experimental activity in the ferroelectrics community. This is mainly due to the challenging experimental requirement of dipole-level resolved electron microscopy characterisation (see e.g. Fig2 of Tian et al Nat Sci Rev 6, 684, 2019) that is likely required to directly image and confirm the existence of such hopfion structures (Scanning probe investigations by comparison are typically less direct and their interpretation is often more subjective and prone to overinterpretation). Furthermore, the fact that Hopfions have been studied theoretically for decades but have never been observed in any solid phase system [Rybakov et al. arXiv:1904.00250 (2019)] (only in magnetic liquids) suggests that they are very challenging to detect.

The following are some further technical points that I would like to raise. The discussion of charge screening effects in the Methods section is a welcome addition and has important implications for the likelihood of experimental detection. However, I would like some clarification on whether authors expect any difference between the scenarios of (i) screening considered as an initial condition (i.e. does the hopfion still form in the presence of screening charge?) or (ii) the case where it is only considered after the hopfion has already formed in the absence of any screening. I would suggest that the authors include a brief summative statement about the role of screening for hopfion stability in the main text. In the supplementary information Fig S4, the authors have presented a model for a 100nm nanosphere and note that each cell/domain has the Hopfion fibration/vortex texture. My question here is whether the model is capable (in principle) of predicting less complex multidomain patterns that do not have the hopfion/vortex characteristic or are there model constraints that prevent this? Also, has the effect of screening been considered for these larger particles? In their rebuttal, the authors' offer some clarifying commentary on the implications of the hopfion switching behaviour for negative capacitance - a brief comment in the main text on the conditions required for observing the negative capacitance contribution would be helpful.

Point-by-point replies to Reviewers

Reviewer #1 (Remarks to the Author):

In the revised versions of the manuscript and supplemental information, the authors have presented additional simulation data as well as a sufficiently detailed description of the employed methods. Overall, I find that the quality of the presentation has significantly improved and that the authors have addressed almost all of my previous comments. In my opinion, the revised version is technically sound and does not raise any major questions.

At the same time, some minor points remain to be clarified. Specifically, I was not convinced by the response of the authors about the topological characterization of the Hopfion state.

Specifically, I suggested that the authors should use Eq. (1) of the manuscript to confirm the Hopfion topology. In response, the authors have indicated that the gauge field A appearing in this formula does not have a physical meaning and that there is no common approach to computing the Hopf index. Why then did the authors provide this formula in the first place if they do not find it useful? At the same time, the Hopf index was already computed by several authors for spin patterns (see e.g. Ref. [28] provided in the manuscript). I do not see any reason why it could not be done for the case of polarization fields discussed in this manuscript. In case the authors do not wish to perform such calculations, I suggest that it is explicitly stated in the main text of the manuscript that the topological characterization is performed through visual inspection of the field patterns and the integer value of the visually assessed topological charge (or linking index) should be provided. The fact that authors did not compute the Hopf index is not a conceptual problem. However, visual inspection can leave certain facts unnoticed and I think it would be better if the method employed by authors to identify the topology of field patterns is stated in an unambiguous manner.

Answer: We thank the Reviewer for this important question since it highlights advances made in our work, based on Arnold's generalization of the Hopf fibrations to the divergenceless vector fields confined in the restricted volume for which the conserving Hopf invariant is not necessarily an integer number.

First and foremost, we investigate Hopfion fibration of the divergenceless field rather than the usually studied Hopfion, for example, in spin patterns, that appears as a trace of the unit vector maintaining its direction during its motion. Hopfion fibrations of the divergenceless field are by far more complicated and less explored, because of the non-local character of the constraint $\text{div}P=0$. Second, we consider the formation of the Hopfion in the restricted confined geometry, i.e. in a simply connected manifold M (rather than in the infinite space R_3) with the field tangent to its boundary. To meet the challenge, we build on Arnold's theorem that discovered and described the corresponding Hopf fibration in the context of the topological hydrodynamics of incompressible liquid. It states that the Hopf invariant H of the divergenceless confined in M field is preserved under the action of an arbitrary volume-preserving diffeomorphism of M , i.e. the system is stable and does not relax out. Importantly, for the confined geometry, the conserving H is not necessarily an integer. The fundamental Equation (1) defining Hopf invariant is useful since it quantifies the stability of the Hopfion: the condition $H \neq 0$ implies the formation of the Hopfion. In order to illustrate that, we utilize another Arnold's theorem generalizing the Gauss linking theorem for the non-integer Hopf invariant at the confined manifolds. Namely, it states that the average self-linking of a divergence-free vector field on a simply connected manifold M coincides with H . The visual inspection of configurations shown in Fig. 2.e straightforwardly reveals the linking of the field lines. Now, although we can calculate the linking number for two field lines (which, for instance, is equal to 2 in the demonstrated configuration) we can not derive an exact, possibly noninteger, value of H , since the necessary averaging over all the pairs of

linked lines of the given field, is an overwhelmingly complex (if possible at all) task. In the revised version of the manuscript, we articulated this point.

We would like to note in this connection that in Ref. [28] of the previous version, the Hopf index was not computed. The authors of [28] used the Hopfion solution for the spin pattern with $H=1$ and $H=2$ obtained for another model in their Ref. [38] and then used this solution as a probe function for the variational procedure and checked that this probe function provides a reasonable minimization for the energy functional describing their spin pattern. Our approach is the opposite: we find numerically the stable configuration of the polarization field and then demonstrate that this solution corresponds to Hopf fibration.

Moreover, I disagree with the statement “To conclude, changing the Hopfion by screening would require an unattainable high free charges density and, therefore, we can neglect screening in our calculations” that the authors have added at the end of the methods section. Specifically, I disagree with the use of the word “unattainable”. Based on the arguments I have provided in the comment about the screening effects I do not think that such a situation is unattainable. The existence of the head-to-head or tail-to-tail charged domain walls is one of the counterexamples. Another specific example (among many) is the distribution of bound charges in polar topological defects found in FE thin films (see Fig. S5(a) in Phys. Rev. Lett. 120,177601 (2018)). In my opinion, this formulation should be changed.

Answer: There is, unfortunately, some lack of clarity concerning the interrelation between the bound charges which indeed, according to the reference provided by the Reviewer, can achieve high values in strained films, and the *free* charges necessary to provide the proper screening effects in Hopfion-hosting nanoparticles. To avoid the confusion we removed the statement in question and gave more details concerning the possible effects of screening in the Main Text and Supplementary Materials, having added new data that we obtained to clarify this point. We thank the Reviewer for this note.

Finally, I would like to point out that in the revised version the authors make a reference to Kolmogorov-Arnold-Moser (KAM) theorem to explain why including elastic and electrostrictive free energy contributions does not alter the Hopfion topology. For me it is not obvious to see why this theorem should apply to the case of static polarization fields considered by the authors. The reference provided by the authors (Arnold’s classical mechanics book) does not help in clarifying the situation since it is very general and provides many different formulations of the theorem. In my opinion the authors need to explicitly demonstrate how exactly the studied case fits either of these formulations.

Answer: We thank the Reviewer for this important point. Indeed, the reference we provided is much too general and refers rather to the application of the KAM theorem to dynamic Hamiltonian systems defined in the even $D=2n$ -dimensional phase space. In fact, the KAM theorem can be generalized for the non-Hamiltonian odd $D=2n+1$ systems with divergenceless (volume-preserving) vector fields, which is our static case for $D=3$. It states that the invariant tori foliation persists under the measure (volume) preserving perturbations i.e. perturbations conserving $\text{div } P = 0$. We added the corresponding comments and references [36,37] into the text.

In conclusion, I would like to mention once again that, despite some issues with specific formulations used by the authors, this study contains rather elegant ideas that would make it interesting not only for researchers working in the fields of ferroelectrics and related materials.

Dr. S. Prokhorenko

Reviewer #2 (Remarks to the Author):

This review is for a revised version of the authors original manuscript previously submitted to Nature Physics. I am pleased to see that the author's revised manuscript contains additional clarifying information in the main text and supplementary information, which helps to give a more complete picture for these new observations, and I think that the work is in a publishable state. In their rebuttal, the author's also make a good case that their first-time prediction of the ferroelectric hopfion will likely draw attention from the theoretical community within ferroelectrics and the wider community working on hopfion theory. Although the study is an interesting and original work, it is my opinion that this study is unlikely to trigger significant additional experimental activity in the ferroelectrics community. This is mainly due to the challenging experimental requirement of dipole-level resolved electron microscopy characterisation (see e.g. Fig2 of Tian et al Nat Sci Rev 6, 684, 2019) that is likely required to directly image and confirm the existence of such hopfion structures (Scanning probe investigations by comparison are typically less direct and their interpretation is often more subjective and prone to overinterpretation). Furthermore, the fact that Hopfions have been studied theoretically for decades but have never been observed in any solid phase system [Rybakov et al. arXiv:1904.00250 (2019)] (only in magnetic liquids) suggests that they are very challenging to detect.

Answer: We are very pleased that Reviewer #2 evaluates highly our revised manuscript and finds it suitable for publication.

The following are some further technical points that I would like to raise.

The discussion of charge screening effects in the Methods section is a welcome addition and has important implications for the likelihood of experimental detection. However, I would like some clarification on whether authors expect any difference between the scenarios of (i) screening considered as an initial condition (i.e. does the hopfion still form in the presence of screening charge?) or (ii) the case where it is only considered after the hopfion has already formed in the absence of any screening.

Answer: Our simulations reveal that the presence of moderate screening slightly modifies a Hopfion structure because screening violates Arnold's theorem. Namely, the fibered nested tori structure distorts, and the tori twinning polarization lines convolve or untwine along the spiral-like paths and creep from torus to torus. The very strong screening unwinds hopfions. This occurs in both scenarios.

I would suggest that the authors include a brief summative statement about the role of screening for hopfion stability in the main text.

Answer: We thank the Reviewer for the suggestion. We added a detailed discussion of screening to the Main Text. The technical details were moved from Methods to the Supplementary Materials and supplementary data requested by the Reviewers were added.

In the supplementary information Fig S4, the authors have presented a model for a 100nm nanosphere and note that each cell/domain has the Hopfion fibration/vortex texture. My question here is whether the model is capable (in principle) of predicting less complex multidomain patterns that do not have the hopfion/vortex characteristic or are there model constraints that prevent this?

Answer: The model that has been used in our calculations is a standard commonly used Ginzburg-Landau-Devonshire model. It does not contain any constraints and yields the structure that minimizes the free energy of the system. Figure S4 indeed presents the result of our calculation evidencing that the polarization lines assume a Hopfion structure. Had any less complex structure possessed the lower energy, the model would have caught such a structure.

Also, has the effect of screening been considered for these larger particles?

Answer: In large particles the consequences of screening are similar. The polarization structure splits into several Hopfions and Hopfioninos, and then each of them is subject to the same fibered nested tori structure distortions, exactly as the Hopfions in smaller particles.

In their rebuttal, the authors' offer some clarifying commentary on the implications of the hopfion switching behaviour for negative capacitance - a brief comment in the main text on the conditions required for observing the negative capacitance contribution would be helpful.

Answer: We thank the Reviewer for the note. We introduced the comment to the text: "This P-E characteristic and, in particular, the negative value of epsilon can be extracted from the total capacitance of the measuring capacitor shown in the inset of Fig. 3. using the Maxwell Garnett mixing rule for the nanoparticle in a dielectric matrix."

REVIEWERS' COMMENTS:

Reviewer #1 (Remarks to the Author):

In their response, the authors have provided sufficient explanations to all my previous questions. Moreover, I appreciate that the authors have further articulated and referenced some of the points pertaining to the topology in the revised version of the manuscript which, in my opinion, would make it easier for the readers to assess the presented arguments. This said, I have no further questions and would like to congratulate the authors on their exciting discovery.

Dr. S. Prokhorenko

Reviewer #2 (Remarks to the Author):

The authors have taken my queries on board, particularly those regarding the role of screening charges on the modelled polarisation textures, and addressed them in the rebuttal to a satisfactory degree. I am also pleased to see that they have taken on my recommendations to include some new commentary in the main text explicitly addressing the role of screening on their models and I think that this strengthens the manuscript. The new figure panels in the supplementary information are also useful in this regard. I have no further queries and believe that this improved manuscript is in a publishable state.